# DenseFace: Bias Mitigation in Face Recognition via Density-Aware Probabilistic Matching

## Abstract

Despite steady progress in face recognition, current face recognition models still suffer from significant demographic biases. While approaches for bias mitigation have been proposed, existing methods often impose constraints on the training procedure and result in the degradation of recognition accuracy. To address this issue, we here introduce a method that reduces demographic biases of pre-trained face recognition models without compromising their accuracy. To this end, we model face embeddings of each person by von Mises-Fisher (MF) distribution. We next observe the dependency between demographic attributes and the density of MF distributions, and propose DenseFace, a probabilistic face matching procedure that accounts for differences in MF distributions. Our extensive experiments demonstrate DenseFace to consistently reduce biases of strong face recognition models varying in network architectures, training datasets and loss functions. Notably, DenseFace preserves accuracy and requires no retraining of existing face recognition models. Our work also investigates previously adopted bias measures and makes suggestions.

## 1 Introduction

Face verification and identification have been significantly improved over the past decade (He et al., 2016; Deng et al., 2019; Schroff et al., 2015; Dan et al., 2023; Shi & Jain, 2019). Nevertheless, current approaches still suffer from demographic biases resulting in unbalanced recognition of people with different gender, race and other attributes (Wang et al., 2019; Conti et al., 2022). For example, in crime investigation, biased predictions may lead to unfair results for people across different demographic groups.

Some of the recent methods for bias mitigation focus on balanced datasets (Wang & Deng, 2020; Sevastopolskiy et al., 2023; Gwilliam et al., 2021). While training on such datasets can reduce recognition bias, limited sizes of such datasets typically imply degradation of recognition accuracy. Other methods address bias reduction with alternative network architectures (Wang et al., 2019; Gong et al., 2021; Huang et al., 2023; Dooley et al., 2024) and loss functions (Wang & Deng, 2020; Xu et al., 2021; Liu et al., 2022; Ma et al., 2023; Kotwal & Marcel, 2024). Such methods, however, typically require retraining followed by the loss of performance (Conti et al., 2022).

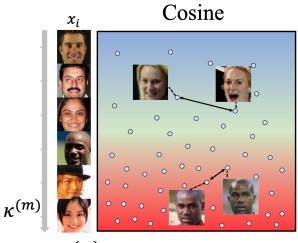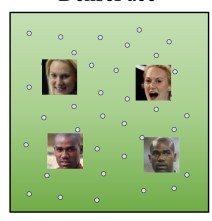

Figure 1: Illustration of bias reduction with DenseFace. (Left): Original face embedding with higher inter-class similarities for Asians and lower inter-class similarities for Caucasians. (Right): DenseFace results in homogeneous inter-class similarities and, in this way, reduces bias for different demographic groups.

Following Terhörst et al. (2020a;b); Dhar et al. (2021); Conti et al. (2022); Linghu et al. (2024) in this work we focus on developing a post-training calibration approach that can be applied to any pre-trained face recognition network. Given face embeddings generated by the target model, we consider each embedding as a sample from von Mises-Fisher (MF) distribution that is known

to be a conditioned Gaussian distribution on a unit hyper-sphere. We then calculate local density of the face embedding space using an anchor set (Liu et al., 2021). We form a subset from the closest embeddings from the anchor set and estimate its local density (Banerjee et al., 2005). We observe the dependency between demographic attributes and the density of MF distributions, and propose *DenseFace*, a probabilistic face matching procedure that accounts for differences in MF distribution. It allows to adjust similarity scores based on face embedding densities as shown in Figure 1. Intuitively, we decrease pairwise face similarities in dense areas of the embedding space and increase similarities in sparse areas. This process corresponds to the expansion and contraction of the embedding space in dense and sparse areas respectively. We also introduce a new margin-based method to compute local densities of nearly orthogonal embeddings.

Our work also revisits the question of measuring face recognition bias. We note that classification accuracy commonly used by the RFW protocol (Wang et al., 2019) may not be suitable in common face recognition scenarios. Moreover, the RFW protocol does not account for cross-racial matching as typically required in real-world scenarios. To address these and other limitations, we adopt the bias evaluation protocol of NIST (Grother et al., 2024) and use false positive rates at a fixed threshold as the primary bias metric, while also assessing the verification accuracy on multi-racial and cross-racial pairs.

Our extensive experiments demonstrate DenseFace to consistently reduce biases of strong face recognition models varying in network architectures, training datasets and loss functions. Notably, DenseFace preserves accuracy and requires no retraining of existing face recognition models.

In summary, we make the following contributions:

- We propose to reduce demographic bias in face recognition with a new density-aware matching procedure based on von Mises-Fisher (MF) distribution. To estimate inter-class densities of face embeddings, we introduce local distortion of embedding spaces and show the importance of using balanced anchor sets.
- Our proposed method *DenseFace* allows to mitigate biases in pre-trained face recognition models without retraining, while preserving recognition accuracy of original models.
- Our experiments demonstrate significant bias reduction for state-of-the-art face recognition models. Following NIST (Grother et al., 2024) evaluation of face verification algorithms, we use false match rate at certain similarity threshold as a more suitable bias evaluation metric.

## 2 RELATED WORK

### 2.1 FACE RECOGNITION

Deep learning has dramatically accelerated the progress in face recognition. The emergence of large-scale face datasets (Nech & Kemelmacher-Shlizerman, 2017; Deng et al., 2019; An et al., 2022; Zhu et al., 2021) with thousands or millions of identities has allowed for training robust and high precision models. State-of-the-art face encoders are usually based on convolutional (He et al., 2016; Deng et al., 2019; Schroff et al., 2015) or attention (Dosovitskiy et al., 2020; Dan et al., 2023) architectures that embed face image into the latent feature space. The choice of a discriminative loss function during training is crucial for an effective faces matching during inference. Having triplet losses (Schroff et al., 2015) at the dawn of deep face recognition, the clear dominance belongs to softmax-based losses (Wang et al., 2017; Liu et al., 2017; Wang et al., 2018) and their extensions (Huang et al., 2020; Kim et al., 2022; Meng et al., 2021) now. These losses incorporate margins into positive or negative class logits of a training sample to increase inter-class separability and decrease intra-class spread.

### 2.2 MEASURING BIAS IN FACE RECOGNITION

To measure bias, Wang et al. (2019) proposed the Racial faces in-the-wild (RFW) testing dataset. Subsequently, in the number of works (Gong et al., 2020; Sevastopolskiy et al., 2023; Gong et al., 2021; Liu et al., 2022; Xu et al., 2021; Ma et al., 2023) model's bias was formulated as the standard deviation of accuracy across four racial subgroups of this test. To complement this, Sevastopolskiy et al. (2023) introduced RB-WebFace with the measure of bias as ROC curve values in each subgroup. Liang et al. (2023) proposed to benchmark model fairness on synthetic dataset with multiple

identities of different races and genders but varying internally in face attributes. Other attempts to measure bias included calculating the geometric mean of FMR and FNMR (Linghu et al., 2024), the skewed error ratio (Wang & Deng, 2020; Kotwal & Marcel, 2024; Huang et al., 2023), the ratio of maximum to minimum values of FMR and FNMR (Conti et al., 2022), and the trade-off between bias reduction and drop in verification performance (Dhar et al., 2021). In this work, we also compare models performance on RFW and RB-WebFace while focusing on FMR values at distance levels fixed across all subgroups. Following NIST (Grother et al., 2024) verification protocol, we argue that this metric better reflects distance dependencies between racial cohorts compared with vanilla accuracy and ROC values.

## 2.3 BIAS MITIGATION IN FACE RECOGNITION

To address model's bias mitigation from the data perspective, Wang & Deng (2020) introduced BUPT-Balancedface dataset with uniform races distribution and BUPT-Globalface with a distribution close to the world's population. Sevastopolskiy et al. (2023) collected unlabeled million-scale African and Asian image sets for self-supervised face encoder pretraining step to ensure equal recognition quality among race groups. Melzi et al. (2024) explored the possibility of bias reduction by introducing the mix of real and synthetic data into training. Zhao et al. (2025) use LLM-guided diffusion to synthesize demography-balanced data and selectively fine-tune bias-sensitive weights (via gradient-difference masks), improving worst-group fairness with minimal accuracy loss. Gwilliam et al. (2021) demonstrated that balanced training datasets not necessarily reduce bias better than skewed ones. We align with the latter conclusion and show that models trained on large scale unbalanced datasets might possess lesser bias. In addition, collecting an annotated balanced dataset is time consuming and challenging.

From the architectural point of view, Wang et al. (2019) introduced information maximization adaptation network to perform domain adaptation from Caucasian group. Gong et al. (2021) proposed an adaptive layer that learns to avoid discrimination among race and gender groups. Huang et al. (2023) proposed to use adversarial learning on sensitive image regions of each racial domain according to gradient attention maps. Dooley et al. (2024) investigated which model design solutions causes the presence of bias and suggested the set of fair backbones found by neural architecture search. In contrast, our method does not require neither face encoder modifications, nor attribute annotation during inference. DenseFace can be directly applied on top of any trained face recognition model.

Some works focused on ensuring fairness by designing loss functions. Wang & Deng (2020) suggested to learn optimal margins for non-Caucasian groups via reinforcement learning. Xu et al. (2021) improved model fairness through FPR penalty loss function. Liu et al. (2022) used sample-level adaptive cosine margins to mitigate the effect of attributes unbalance in training data. Ma et al. (2023) proposed to keep face embedding invariant across demographic attribute groups via self-supervised data partitioning learning strategy. Kotwal & Marcel (2024) induced score calibration into training by adding regularization loss based on races scores misalignment. Unlike this branch of works, our DenseFace is designed to mitigate bias of a face recognition model without any need for re-training it.

Finally, the problem of acquiring fairness can be addressed with adjusting the decision rule instead of fine-tuning the learned latent space. Terhörst et al. (2020a) suggested to assign unique distance thresholds according to face clustering results. In another work (Terhörst et al., 2020b), score distributions of different racial domains were made to be similar by an additional classifier that replaced similarity function. Dhar et al. (2021) introduced an adversarial technique on top of a trained model to eliminate sensual attribute information from face descriptor. Conti et al. (2022) relied on minimizing von Mises-Fisher loss to project an embedding of a trained model into more fair latent space. Linghu et al. (2024) proposed score normalization methods that reduce ethnicity and gender bias. However, score calibration and embedding post-processing usually introduce a trade-off between eliminating bias and preserving verification metrics. In contrast to previous methods, our proposed DenseFace does not downgrade verification performance while significantly reducing bias.

## 3 METHODOLOGY

Our method is based on the idea that face embedding densities contain demographic information. It consists of three stages. Using these densities, we can modify matching algorithm to mitigate bias.

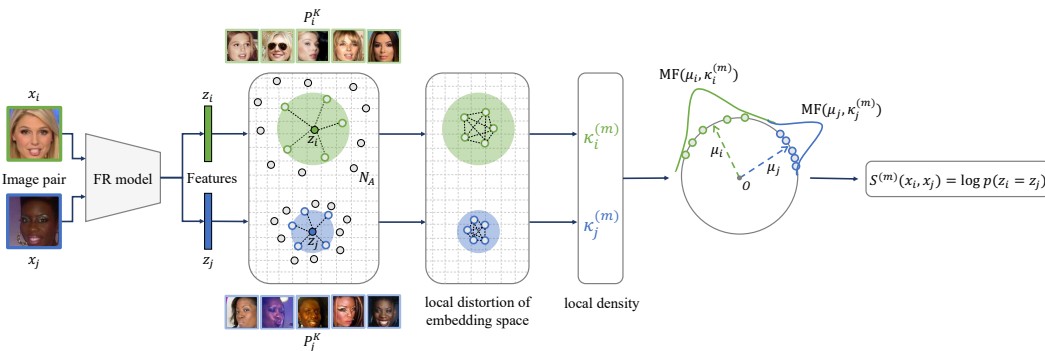

Figure 2: Pipeline of density-aware probabilistic matching. Face recognition model takes two input images $x_i$ and $x_j$ for verification task and generates embeddings $z_i$ and $z_j$. $K$ nearest neighbors sets $P_i^K$ and $P_j^K$ for face embeddings $z_i$ and $z_j$ are obtained from anchor set $N_A$ (Section 3.1). Then, we locally distort embedding space by applying margin $m$ in Equation (3) to estimate local densities $\kappa_i^{(m)}$ and $\kappa_j^{(m)}$ of the $P_i^K$ and $P_j^K$ (Section 3.2). Bigger radius of colored circle around the face embedding corresponds to smaller local density. Probabilistic embedding representations $p(z|\mu_i, \kappa_i^{(m)})$ and $p(z|\mu_j, \kappa_j^{(m)})$ are then constructed as MF distributions with $\mu_i = z_i$ and $\mu_j = z_j$. Mutual likelihood of these representations to belong to the same identity is considered as matching score for face verification (Section 3.3).

Firstly, we utilize pre-trained face recognition model to estimate probabilistic face embeddings. Secondly, we introduce local distortion of embedding space that allows to calculate local densities of face embbedings. Finally, we perform matching of two probabilistic face embeddings based on likelihood of their distributions to be associated with the same person. Whole pipeline of DenseFace is shown in Figure 2.

## 3.1 PROBABILISTIC FACE EMBEDDING REPRESENTATION

Having pre-trained neural network $Z(x)$, we consider face embedding $z_i = Z(x_i)$ of the face image $x_i$ as probabilistic distribution $p(z)$ in the feature space. As state-of-the-art $d$-dimensional face embeddings are usually trained with assumption that during inference they are set on a unit hyper-sphere $\mathbb{S}^{d-1}$ (Deng et al., 2019), the MF distribution is a reasonable choice to represent face embedding (Banerjee et al., 2005; Conti et al., 2022; Li et al., 2021; Nech & Kemelmacher-Shlizerman, 2017). The probability density of MF distribution with mean direction $\mu_i \in \mathbb{S}^{d-1}$ and density parameter $\kappa_i > 0$ is defined as follows:

$$p(z|\mu_i, \kappa_i) = C_d(\kappa_i)e^{\kappa_i \mu_i^T z} \tag{1}$$

$$C_d(\kappa_i) = \frac{\kappa_i^{\frac{d}{2}-1}}{(2\pi)^{d/2} I_{\frac{d}{2}-1}(\kappa_i)} \tag{2}$$

$I_{\frac{d}{2}-1}$ stands for the modified Bessel function of the first kind at order $\frac{d}{2} - 1$. We follow Banerjee et al. (2005) to estimate $\kappa_i$ values:

$$\kappa_i = \frac{r_i(d - r_i^2)}{1 - r_i^2} \tag{3}$$

$$r_i = \frac{\left\| \sum_{k=1}^{K} \xi_i^k \right\|}{K} \tag{4}$$

where $\xi_k^i$ are the face embeddings belonging to this distribution and $\|\cdot\|$ stands for $l_2$-norm. Figure 1 illustrates estimated density parameters $\kappa$ of the MF distribution for face embeddings - it is higher in regions with high density of face embeddings in the feature space. In practice, to estimate $p(z|\mu_i, \kappa_i)$ we follow idea from Liu et al. (2021) and use additional embedding anchor set $N_A$ obtained from image anchor set $X_A$. We set $\mu_i$ equal to the face embedding $z_i$. For $\kappa_i$ calculation we find $K$ nearest neighbors $\xi_i^k \in N_A, k = 1 \ldots K$ of the face embedding $z_i$. We call this density, evaluated based on the embedding anchor set, as local density. Nearest neighbors embeddings form set $P_i^K \subseteq N_A$.

## 3.2 LOCAL DISTORTION OF EMBEDDING SPACE

Previously, similar approach for face embedding density calculation based on MF distribution was used in Oinar et al. (2023), where face embeddings of the same identity were used as the embedding anchor set. However, we found that densities computed based on the intra-class distribution of face embeddings do not really represent racial bias (see Figure 3). For example, if we consider class with near similar face images (i.e., obtained from consequen-

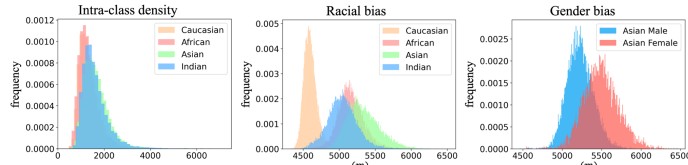

Figure 3: Distributions of intra-class densities $\kappa$ of RB-WebFace embeddings for different racial groups (left). These distributions are quite similar for all racial groups and do not represent racial bias of face embeddings. Distributions of inter-class local densities $\kappa^{(m)}$ of RB-WebFace embeddings for different racial groups (middle) and genders (right). These distributions represent racial and gender bias of face embeddings.

tial frames of some video) it will have extremely high density no matter which gender or race does the person form this class belong to. Thus, in our method we mainly focus on inter-class face embedding density. We set anchor set embeddings from $N_A$ as mean representation of separate identities.

For set of embeddings $\xi_i^k \in P_i^K$ its sum in Equation (4) can be rewritten as

$$\left\| \sum_{k=1}^{K} \xi_i^k \right\| = \sqrt{\sum_{k=1}^{K} \|\xi_i^k\|^2 + 2\sum_{k \neq l} \|\xi_i^k\|\|\xi_i^l\| \cos \Theta_i^{kl}} \tag{5}$$

where $\cos \Theta_i^{kl}$ are pairwise cosines between $\xi_i^k, \xi_i^l \in P_i^K$. In practice, when we calculate densities $\kappa_i$, there is a problem of near-similar values of $\kappa_i$ for all face embeddings because of near-orthogonality of all of the embeddings from the anchor set: all $\cos \Theta_i^{kl}$ are concentrated near zero value and $\left\| \sum_{k=1}^{K} \xi_i^k \right\| \simeq \sqrt{\sum_{k=1}^{K} \|\xi_i^k\|^2} = \sqrt{K}$. To fix this issue, we introduce angular margin $m$ inside the cosine function:

$$\left\| \sum_{k=1}^{K} \xi_i^k \right\|^{(m)} := \sqrt{\sum_{k=1}^{K} \|\xi_i^k\|^2 + 2\sum_{k \neq l} \|\xi_i^k\|\|\xi_i^l\| f(\Theta_i^{kl})} \tag{6}$$

$$f(\Theta_i^{kl}) = \begin{cases} \cos(\Theta_i^{kl} - m) & \text{if } \Theta_i^{kl} > m \\ 1 & \text{otherwise.} \end{cases} \tag{7}$$

As a result, the distribution of $\cos \Theta_i^{kl}$ shifts to higher values in Figure 4. Density value computed with margin $m$ is denoted as $\kappa_i^{(m)}$. Distribution of $\kappa_i^{(m)}$ for different racial groups and genders is shown in Figure 3. It can be seen that racial and gender biases are now encoded in densities: faces of non-Caucasian and female identities obtain higher densities, which correlates with the fact that they are harder to recognize for state-of-the-art face recognition models. We call the procedure of margin insertion as local distortion of embedding space, because it effects like local squeezing leading to higher cosine similarities between embeddings. Local densi-

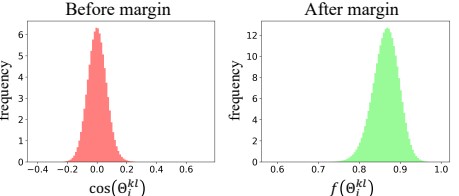

Figure 4: Distributions of pairwise cosine similarities $\cos(\Theta_i^{kl})$ between anchor set embeddings on the left and $f(\Theta_i^{kl})$ from Equation (7) on the right.

ties $\kappa^{(m)}$ can further be learned to accelerate embedding matching and avoid usage of anchor set during inference (see (Section 4.2) for implementation details).

### 3.3 DENSITY-AWARE PROBABILISTIC MATCHING

During matching, we need to calculate similarity score between two face images $x_i$ and $x_j$. For most of the state-of-the-art face recognition models classic approach of cosine similarity $s_{ij} = \cos(z_i^T z_j)$ between face embeddings $z_i, z_j$ is used.

For probabilistic embedding matching we use mutual likelihood of two probability distributions $p(z|\mu_i, \kappa_i)$ and $p(z'|\mu_j, \kappa_j)$ to belong to the same person as a matching score:

$$S_{ij} = \log \iint_{\mathbb{S}^{d-1} \times \mathbb{S}^{d-1}} p(z|\mu_i, \kappa_i) p(z'|\mu_j, \kappa_j) \delta(z - z') dz dz' \tag{8}$$

We follow Li et al. (2021) for mutual likelihood estimation of two MF distributions on unit hypershere and use $\kappa^{(m)}$ as local densities:

$$S_{ij}^{(m)} = \log C_d(\kappa_i^{(m)}) + \log C_d(\kappa_j^{(m)}) - \log C_d(\|\kappa_i^{(m)}\mu_i + \kappa_j^{(m)}\mu_j\|) \tag{9}$$

Overall, we call our density-aware probabilistic matching as DenseFace. Final scheme of our method is shown in Figure 2. Algorithm 1 describes whole DenseFace pipeline.

---

**Algorithm 1:** DenseFace algorithm

---

**Data:** Images $x_i$ and $x_j$, image anchor set $X_A$, face recognition network $Z(x)$.

**Result:** $S_{ij}^{(m)}$ - similarity score between $x_i$ and $x_j$.

1. Compute face embeddings: $z_i = Z(x_i), z_j = Z(x_j)$

2. Compute embedding anchor set $N_A$: $N_A = \{Z(x_m), x_m \in X_A\}$

3. Compute nearest neighbors embedding sets $P_i^K$ and $P_j^K$.

4. Estimate local densities $\kappa_i^{(m)}$ and $\kappa_j^{(m)}$ with Equation (3) and Equation (6).

5. Compute $S_{ij}^{(m)}$ with Equation (9).

---

## 4 EXPERIMENTS

### 4.1 DATASETS

We evaluate our method on RFW (Wang et al., 2019) and RB-WebFace (Sevastopolskiy et al., 2023) benchmarks. For building anchor image set and training our learning-based approach, we utilize Glint360K (An et al., 2022) dataset. We consider models trained on MS1MV2 (Deng et al., 2019), Glint360K, WebFace4M, and WebFace12M (Zhu et al., 2021) datasets.

### 4.2 IMPLEMENTATION DETAILS

We follow Wang et al. (2018); Deng et al. (2019) by aligning and cropping all face images to $112 \times 112$ resolution with five landmarks provided by MTCNN (Zhang et al., 2016).

We employ modified versions of ResNet-50 and ResNet-100 (Deng et al., 2019) to extract 512-dimensional face embeddings.

For balanced image anchor set $X_A$ construction, we adopt pre-trained gender and race classifiers, that estimate these attributes of faces in Glint360K dataset. Then, we randomly select 54,000 identities from Glint360K such that the numbers of IDs in subgroups defined by one of the four race groups and gender are equal. We construct embedding anchor set by calculating the average embedding for each identity. The size of the nearest neighbors embedding set $K$ is 128.

For learning based approach, we train $\kappa^{(m)}$ regression network consisting of two fully-connected layers, ReLU activation functions and batch normalization layers to minimize MSE loss function. The hidden dimension is equal to 256. Models are trained on Glint360K using SGD optimizer with the weight decay of 2e-3 for 100 epochs. Initial learning rate is set to 0.1 and decreased by cosine annealing scheduler. The batch size is set to 2048.

Table 1: Comparison of the verification accuracy (%) of the methods on RFW validation set. Upper half belongs to models trained on BUPT-Balancedface. Lower half belongs to models trained on large unbalanced datasets. Asterisk "*" indicates that the results are directly taken from the corresponding paper.

| Model | Cauc. | Afr. | Asian | Ind. | Avg (↑) | Std (↓) |
|---|---|---|---|---|---|---|
| RL-RBN-R34* (Wang & Deng, 2020) | 96.27 | 95.00 | 94.82 | 94.68 | 95.19 | 0.63 |
| DebFace-R34* (Gong et al., 2020) | 95.95 | 93.67 | 94.33 | 94.78 | 94.68 | 0.83 |
| DAM-R34* (Liu et al., 2021) | 96.30 | 94.51 | 94.31 | 95.20 | 95.08 | 0.78 |
| ArcFace-R50* (Xu et al., 2021) | 95.55 | 94.95 | 96.68 | 95.47 | 95.66 | 0.63 |
| CIFP-R50* (Xu et al., 2021) | 97.08 | 96.47 | 95.75 | 96.77 | 96.52 | 0.49 |
| GAC-R50* (Gong et al., 2021) | 96.27 | 94.40 | 94.32 | 94.77 | 94.94 | 0.79 |
| StyleGAN-R50* (Sevastopolskiy et al., 2023) | 96.52 | 95.00 | 93.90 | 94.93 | 95.09 | 0.94 |
| ArcFace-R100* (Xu et al., 2021) | 96.43 | 94.98 | 97.37 | 96.17 | 96.24 | 0.85 |
| CIFP-R100* (Xu et al., 2021) | 97.03 | 95.65 | 97.60 | 96.82 | 96.78 | 0.71 |
| CosFace-R50-Glink360K (An et al., 2022) | 98.50 | 98.08 | 99.48 | 98.38 | 98.61 | 0.53 |
| AdaFace-R50-WebFace4M (Kim et al., 2022) | 97.52 | 96.68 | 98.57 | 97.53 | 97.57 | 0.67 |
| AdaFace-R100-WebFace4M (Kim et al., 2022) | 98.42 | 97.80 | 99.47 | 98.12 | 98.45 | 0.63 |
| AdaFace-R100-WebFace12M (Kim et al., 2022) | 98.87 | 98.55 | 99.43 | 98.67 | 98.88 | 0.34 |

## 4.3 PERFORMANCE METRICS

We argue that common approaches to measure ethnic bias adopted in previous works (Gong et al., 2020; Sevastopolskiy et al., 2023; Gong et al, 2021; Liu et al., 2022; Xu et al., 2021; Ma et al., 2023) do not fully represent model "biasness" for real-world applications.

For instance, the most famous RFW (Wang et al., 2019) protocol suggests reporting verification accuracy for four ethnicity groups with predefined face image pairs. Optimal distance threshold is defined separately for each race by k-fold cross validation approach. A method is said to decrease model's bias if its standard deviation (Std) of accuracy across groups is decreased. However, real biometric systems rarely can afford to set a unique threshold for each racial domain. Moreover, we show that Std is inconsistent as the measure of bias on the example of two models. As shown in Table 1, CosFace-R50-Glink360K has lower Std than AdaFace-R50-WebFace4M. However, the difference between mean cosine similarities of RFW negative pairs for Caucasian group and other races reveals that the latter model is less biased (see Figure 5).

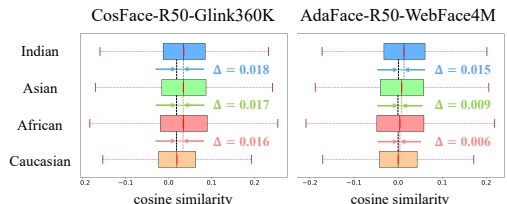

Figure 5: Cosine similarities for negative pairs of RFW test for each racial group. The difference between mean cosine similarities for Caucasian group and other races is denoted as Δ. According to Table 1 CosFace-R50-Glink360K has lower racial bias than AdaFace-R50-WebFace4M. However, it has larger Δ for all racial subgroups which indicates higher racial bias.

RB-WebFace (Sevastopolskiy et al., 2023) benchmark consists of positive and negative sets of images assembled from WebFace42M dataset. The test includes reporting ROC curves values, namely TPR at FPR = $\{10^{-3}, 10^{-4}\}$, independently for four racial groups. The metrics value is approximated by varying the similarity threshold in a predefined range of values. Although ROC curve consideration eliminates the need to manually set a threshold, RB-WebFace protocol does not reveal that a biased model has different similarity values across ethnicity groups at the same level of FPR, which likewise affects production scenarios.

To address these limitations, we propose to measure ethnicity bias on these two tests according to NIST report on verification (Grother et al., 2024) as FPR measurements at the value of similarity that gives FPR of Caucasian group (analogous to "MW" in NIST report) equal to 0.001. Reducing FPR below $10^{-3}$ for some groups may lead to the presence of bias on the Caucacisan domain. This metric allows to track not only verification performance of individual groups but also consistency of similarity ranges among them. In contrast to RB-WebFace protocol, we calculate TPR at FPR having all unique similarity values as thresholds without any predefined range which allows for more precise metric estimation. Instead of using predefined test pairs, we consider all possible pairs of images in RFW. This results in approximately 14 thousands of positive and 50 millions of negative pairs per group compared to 3000 positive and 3000 negative pairs in vanilla RFW. The original pairs of RB-WebFace are preserved.

## 4.4 DEBIAS OF SOTA MODELS

Table 2: RFW, NIST protocol, FPR @ similarity (Caucasian FPR = $10^{-3}$), $\times 10^3$ scale, the closer to 1 the better

| | CosFace R50 Glink360K | | AdaFace R50 WebFace4M | | AdaFace R100 WebFace4M | | AdaFace R100 WebFace12M | |
|---|---|---|---|---|---|---|---|---|
| | Cosine | DenseFace | Cosine | DenseFace | Cosine | DenseFace | Cosine | DenseFace |
| Caucasian | 1.00 | 1.00 | 1.00 | 1.00 | 1.00 | 1.00 | 1.00 | 1.00 |
| African | 8.82 | **0.91** | 7.67 | **1.10** | 7.98 | **1.22** | 6.66 | **1.71** |
| Asian | 7.69 | **0.43** | 5.84 | **0.63** | 5.55 | **0.81** | 4.77 | **1.19** |
| Indian | 5.96 | **0.62** | 5.62 | **0.73** | 5.93 | **0.89** | 5.44 | **1.36** |

In this work, we examine strong open-source AdaFace (Kim et al., 2022) and CosFace (An et al., 2022) models. These state-of-the-art models were trained on datasets (MS1MV2, Glint360K, WebFace4M, and WebFace12M) with non-uniform ethnicities distribution which is close to real training regimes. Following Dooley et al. (2024), we question the necessity of preserving a demographic balance in a training dataset. Unlike previous works, we demonstrate that training on large-scale skewed datasets possess lesser or comparable bias according to Std on RFW in comparison to models and debiasing methods trained on BUPT-Balancedface (see Table 1).

To avoid leaks between train and test, we evaluate models trained on WebFace4M and WebFace12M only on RFW. On RB-WebFace, we evaluate models trained on MS1MV2 and Glint360K.

As shown in Table 2 and Table 3, applying our DenseFace significantly reduces racial bias for all non-Caucasian domains compared with the cosine similarity baseline, while preserving or improving its verification performance (see Table 4).

Table 3: RB-WebFace, NIST protocol, FPR @ similarity (Caucasian FPR = $10^{-3}$), $\times 10^3$ scale, the closer to 1 the better

| | AdaFace R50 MS1MV2 | | AdaFace R100 MS1MV2 | | CosFace R50 Glink360K | |
|---|---|---|---|---|---|---|
| | Cosine | DenseFace | Cosine | DenseFace | Cosine | DenseFace |
| Caucasian | 1.00 | 1.00 | 1.00 | 1.00 | 1.00 | 1.00 |
| African | 2.69 | **0.80** | 2.86 | **1.39** | 3.74 | **0.40** |
| Asian | 5.16 | **2.38** | 5.34 | **3.07** | 9.96 | **0.71** |
| Indian | 4.50 | **1.73** | 4.74 | **1.75** | 5.19 | **0.92** |

Table 4: RFW and RB-WebFace, TPR @ FPR (%)

| Method | Cauc. | Afr. | Asian | Ind. | Avg (↑) | Cauc. | Afr. | Asian | Ind. | Avg (↑) |
|---|---|---|---|---|---|---|---|---|---|---|
| TPR @ FPR=$10^{-3}$ | AdaFace-R100-WebFace12M, RFW | | | | | AdaFace-R50-MS1MV2, RB-WebFace | | | | |
| Cosine | 99.82 | 99.62 | 99.41 | 99.51 | **99.59** | 97.54 | 94.63 | 98.36 | 98.41 | 97.24 |
| DAM (Liu et al., 2021) | 99.81 | 99.59 | 99.38 | 99.49 | 99.57 | 97.63 | 94.80 | 98.32 | 98.39 | 97.28 |
| DenseFace | 99.84 | 99.62 | 99.32 | 99.58 | **99.59** | 97.57 | 94.94 | 98.20 | 98.52 | **97.31** |
| TPR @ FPR=$10^{-4}$ | AdaFace-R100-WebFace12M, RFW | | | | | AdaFace-R50-MS1MV2, RB-WebFace | | | | |
| Cosine | 99.49 | 98.95 | 97.98 | 98.51 | 98.73 | 94.78 | 89.62 | 96.06 | 96.04 | 94.12 |
| DAM (Liu et al., 2021) | 99.47 | 98.96 | 97.83 | 98.49 | 98.69 | 94.92 | 89.91 | 96.16 | 96.02 | 94.25 |
| DenseFace | 99.50 | 99.03 | 97.92 | 98.60 | **98.76** | 94.83 | 90.10 | 95.89 | 96.27 | **94.27** |

## 4.5 LEARNING-BASED APPROACH

In this subsection, we present the results of training $\kappa^{(m)}$ regression network for faster matching inference speed. In Table 5 we provide results for AdaFace-R100-WebFace12M and AdaFace-R50-MS1MV2 on RFW and RB-Webface respectively. Surprisingly, learning-based DenseFace[†] achieves even better results in terms of bias mitigation than anchor set based DenseFace for both datasets.

Table 5: Learning-based approach, RFW and RB-WebFace, NIST protocol, FPR @ similarity (Caucasian FPR = $10^{-3}$), $\times 10^3$ scale, the closer to 1 the better

| Methods | Cauc. | Afr. | Asian | Ind. | Cauc. | Afr. | Asian | Ind. |
|---|---|---|---|---|---|---|---|---|
| | RFW AdaFace-R100 WebFace12M | | | | RB-WebFace AdaFace-R50 MS1MV2 | | | |
| Cosine | 1.00 | 6.66 | 4.77 | 5.44 | 1.00 | 2.69 | 5.16 | 4.50 |
| DenseFace | 1.00 | 1.71 | 1.19 | 1.36 | 1.00 | **0.80** | 2.38 | 1.73 |
| DenseFace[†] | 1.00 | **1.38** | **1.01** | **1.23** | 1.00 | 0.62 | **1.87** | **1.51** |

## 4.6 QUANTITATIVE ANALYSIS

From Figure 3, we make an observation that large values of density $\kappa^{(m)}$ on non-Caucasian domains correspond to high values of FPR for these subgroups (see Table 2 and Table 3). This highlights the possibility of using $\kappa^{(m)}$ value as an indicator of racial bias presence.

In Figure 6, we provide the distribution of races among $K$ nearest neighbors for each of the RFW racial subgroups. These statistics demonstrate that for each race the majority of neighbors belongs to the same race. Therefore, anchor set can be considered as an implicit race classifier. Due to this property, our DenseFace does not require ground truth racial labels during inference. Inducing the demographic balance into the image anchor set allows each race to find the sufficient number of same race

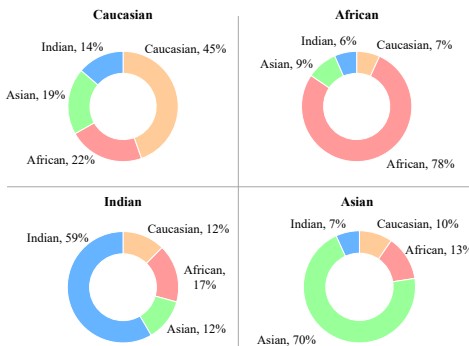

Figure 6: Race distribution among $K$ nearest neighbors from feature anchor set for each race on RFW test.

neighbors in it. The presence of other races in nearest neighbors distributions can be explained by the cross-racial effect of bias from other attributes like gender or age.

### 4.7 QUALITATIVE ANALYSIS

On the example of face images $x_i$ with low and high $\kappa^{(m)}$ values, we visualize their 10 nearest neighbors from the image anchor set on Figure 7. Noticeably, these neighbors not only belong to the same race, but also possess some of the same face attributes like hair style and facial expression, which may have a positive impact on other types of bias during DenseFace inference. In addition, the ranking of face images $x_i$ according to $\kappa^{(m)}$ aligns with our findings that non-Caucasian and female identities have higher local density values in comparison to Caucasian male group (see Figure 3).

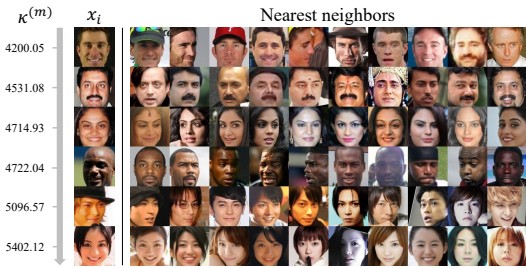

Figure 7: Examples of images with low and high $\kappa^{(m)}$ and their 10 nearest neighbors from the image anchor set.

### 4.8 COMPUTATIONAL COST ANALYSIS AND LIMITATIONS

We present a dedicated study of the runtime and memory requirements in Appendix G of the supplementary. In particular, Table 16 summarises the latency and memory requirements for the full pipeline of backbone inference and local-density estimation, excluding pairwise face matching. The learning based version of our method (DenseFace$^{\dagger}$) implies only marginal increase in memory (+0.2% or +132k parameters) and latency (+1.75% or +131k MACs) compared to the baseline. Table 16 further reports times for computing pairwise face similarities $S^{(m)}$ according to Equation (9). We observe that direct application of Equation (9), as well as SCF probabilistic matching in Li et al. (2021), implies 2.5 slower face matching using DenseFace compared to the Cosine baseline. The computational cost of $S^{(m)}$, however, is dominated by the calculation of modified Bessel functions in Equation (2). Hence, we use lookup tables and JIT optimisation, and reduce computational cost for $S^{(m)}$ from 0.071ms to 0.042ms. While DenseFace matching remains 1.5 slower compared to Cosine, the overall complexity of DenseFace should be acceptable for the majority of practical applications and can be improved with further numerical optimizations.

## 5 CONCLUSION

In this paper, we have investigated the racial bias of state-of-the-art face recognition models. We have introduced DenseFace - new probabilistic matching method of face embeddings that mitigates racial bias by incorporating embedding densities. Extensive experiments among different neural network architectures, training datasets and loss functions demonstrate the effectiveness of our proposed DenseFace without re-training of face recognition models.

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
