This Supplementary Material provides additional details for the DenseFace approach. We present training details for BUPT-BalancedFace in Section A, additional experimental comparisons in Section B, evaluation on conventional face benchmarks in Section C, evaluation results according to RFW protocol in Section D, ablation studi in Section E, face verification results on multi-racial dataset in Section F, limitations of the proposed approach in Section G as well as additional details on the anchor set and face image filtering in Sections H and I respectively.

## A   TRAINING ON BUPT-BALANCEDFACE

Following common de-biasing research setup, we also train ResNet-50 on BUPT-Balancedface Wang & Deng (2020). Results are presented in Table 6. Despite of training on the balanced dataset, ResNet-50 model trained with CosFace is highly biased. DenseFace mitigates bias for African and Indian domains: it reduces FPR at certain threshold in 4 times from 0.00615 to 0.00143 and from 0.00401 to 0.00103 correspondingly.

Table 6: RFW, training on BUPT-Balancedface, NIST protocol, FPR @ similarity (Caucasian FPR $= 10^{-3}$), $\times 10^3$ scale, the closer to 1 the better

| Methods | Cauc. | Afr. | Asian | Ind. |
|---|---|---|---|---|
| CosFace-R50-BUPT-Balancedface | | | | |
| Cosine | 1.00 | 6.15 | 1.66 | 4.01 |
| DenseFace | 1.00 | **1.43** | **0.88** | **1.03** |

## B   COMPARISON WITH OTHER DE-BIASING METHODS

To demonstrate remaining racial bias in models trained with state-of-the-art de-biasing approaches, we evaluated available open-source de-biased models on RFW benchmark with FPR at similarity metric. Althought ResNet-34 model trained with CIFP Xu et al. (2021) approach on BUPT-Balancedface has less bias than CosFace-R34-Glint360K, it still has huge difference of FPR among different racial groups (Table 7). However, applying of DenseFace allows to mitigate remaining bias.

Table 7: RFW, NIST protocol, FPR @ similarity (Caucasian FPR $= 10^{-3}$), $\times 10^3$ scale, the closer to 1 the better

| Methods | Cauc. | Afr. | Asian | Ind. | Cauc. | Afr. | Asian | Ind. |
|---|---|---|---|---|---|---|---|---|
| | CIFP-R34 Xu et al. (2021) | | | | CosFace-R34-Glint360K | | | |
| Cosine | 1.00 | 5.43 | 1.54 | 3.21 | 1.00 | 9.35 | 7.58 | 5.67 |
| DenseFace | 1.00 | **1.70** | **0.81** | **0.80** | 1.00 | **1.10** | **0.46** | **0.73** |

We also acknowledge the work of Gong et al. (2020; 2021); Huang et al. (2023); Liu et al. (2022); Sevastopolskiy et al. (2023); Wang & Deng (2020), but do not compare with them directly as the code, the pretrained models, and the results for our experimental setup were not available for these methods. We reimplemented Conti et al. (2022); Linghu et al. (2024) and evaluated these methods in the AdaFace-R100-WebFace12M setup of Table 5. The results reported in Table 8 indicate significant improvements of our method.

## C   EVALUATION ON STANDARD FACE RECOGNITION BENCHMARKS

We have performed evaluations for the strongest AdaFace-R100-WebFace12M model on standard benchmarks LFW Huang et al. (2008), CFP-FP Sengupta et al. (2016), AgeDB Moschoglou et al. (2017), CPLFW Zheng & Deng (2018), CALFW Zheng et al. (2017), and IJB-C Maze et al. (2018). Along with Table 4 these results (Table 9) demonstrate that DenseFace allows to maintain original accuracy on standard benchmarks while significantly improving the baseline for the racially representative test set (see Table 14).

Table 8: RB-WebFace, NIST protocol, FPR @ similarity (Caucasian FPR = $10^{-3}$), $\times 10^3$ scale, the closer to 1 the better

| Methods | Cauc. | Afr. | Asian | Ind. |
|---|---|---|---|---|
| AdaFace-R100-WebFace12M | | | | |
| EM-FAR Conti et al. (2022) | 1.00 | 6.56 | 4.80 | 6.15 |
| SN-M3 Linghu et al. (2024) | 1.00 | 4.03 | 2.07 | 2.70 |
| Cosine | 1.00 | 6.66 | 4.77 | 5.44 |
| DenseFace | 1.00 | 1.71 | 1.19 | 1.36 |
| DenseFace$^\dagger$ | 1.00 | **1.38** | **1.01** | **1.23** |

Table 9: Verification accuracy (%) for DenseFace on LFW, CFP-FP, AgeDB, CPLFW, CALFW. TPR@FPR=0.01% for IJB-C

| Methods | LFW | CFP-FP | AgeDB | CPLFW | CALFW | IJBC |
|---|---|---|---|---|---|---|
| AdaFace-R100-WebFace12M | | | | | | |
| Cosine | 99.78 | **99.13** | 98.02 | 94.30 | 95.97 | 97.66 |
| DenseFace | 99.78 | 99.01 | 98.08 | **94.32** | **96.10** | 97.59 |
| DenseFace$^\dagger$ | **99.83** | 98.96 | **98.12** | 94.23 | 96.03 | **97.73** |

## D DENSEFACE ON ORIGINAL RFW PROTOCOL

We also evaluate DenseFace on the standard RFW protocol. The results of state-of-the-art models trained on Glint360K and the subsets of WebFace42M are shown in Table 10. DenseFace approach slightly reduces bias in terms of the standard deviation of the accuracy metric, without reduction of the average accuracy.

Table 10: Verification accuracy (%) for DenseFace on RFW validation set. R50 is trained with CosFace, R100 is trained with AdaFace

| Model | Matching | Cauc. | Afr. | Asian | Ind. | Avg (↑) | Std (↓) |
|---|---|---|---|---|---|---|---|
| R50-Glink360K An et al. (2022) | Cosine | 98.50 | 98.08 | 99.48 | 98.38 | 98.61 | 0.53 |
| | DenseFace | 98.55 | 98.33 | 99.43 | 98.45 | 98.68 | 0.44 |
| R100-WebFace4M Kim et al. (2022) | Cosine | 98.42 | 97.80 | 99.47 | 98.12 | 98.45 | 0.63 |
| | DenseFace | 98.55 | 97.78 | 99.43 | 98.13 | 98.48 | 0.62 |
| R100-WebFace12M Kim et al. (2022) | Cosine | 98.87 | 98.55 | 99.43 | 98.67 | 98.88 | 0.34 |
| | DenseFace | 99.02 | 98.53 | 99.35 | 98.65 | 98.89 | 0.32 |

## E ABLATION STUDY

Table 11: Ablation study on matching, RFW, NIST protocol, FPR @ similarity (Caucasian FPR = $10^{-3}$), $\times 10^3$ scale, the closer to 1 the better. $\overline{\text{DAM}}$ denotes DAM with feature anchor set composed from averaged embeddings of the identity. $\overline{\text{DAM}}$ ☮ denotes $\overline{\text{DAM}}$ with race and gender balance in anchor set.

| Methods | Cauc. | Afr. | Asian | Ind. | Cauc. | Afr. | Asian | Ind. |
|---|---|---|---|---|---|---|---|---|
| | CosFace-R50-Glint360K | | | | AdaFace-R50-WebFace4M | | | |
| Cosine | 1.00 | 8.82 | 7.69 | 5.96 | 1.00 | 7.67 | 5.84 | 5.62 |
| DAM (Liu et al., 2021) | 1.00 | 8.97 | 7.49 | 6.52 | 1.00 | 7.02 | 5.65 | 6.00 |
| $\overline{\text{DAM}}$ | 1.00 | 8.02 | 6.62 | 6.06 | 1.00 | 6.29 | 4.91 | 5.40 |
| $\overline{\text{DAM}}$ ☮ | 1.00 | 0.82 | **0.67** | 0.56 | 1.00 | 0.72 | 0.57 | 0.56 |
| DenseFace | 1.00 | **0.91** | 0.43 | **0.62** | 1.00 | **1.10** | 0.63 | **0.73** |

In this subsection, we present ablation study of our method. In Table 11 we start from baseline cosine similarity matching and then add our DenseFace matching algorithm step-by-step. Firstly, we enable anchor set from DAM (Liu et al., 2021) and observe that it does not mitigate bias in terms of FPR @ similarity. Then, we replace embeddings of the anchor set with mean representation of face identities ($\overline{\text{DAM}}$) and slightly reduce bias. Then, we induce racial balance ($\overline{\text{DAM}} \, \circledcirc$) in image anchor set $X_A$ and mitigate bias drastically. Finally, we apply our probabilistic matching DenseFace and obtain the best results.

Table 12: Ablation study on nearest neighbors number $K$, RFW, NIST protocol, FPR @ similarity (Caucasian FPR = $10^{-3}$), $\times 10^3$ scale, the closer to 1 the better

| $K$ | Cauc. | Afr. | Asian | Ind. | Cauc. | Afr. | Asian | Ind. |
|---|---|---|---|---|---|---|---|---|
| | CosFace-R50-Glint360K | | | | AdaFace-R50-WebFace4M | | | |
| 10 | 1.00 | 0.03 | 0.01 | 0.02 | 1.00 | 0.03 | 0.02 | 0.03 |
| 32 | 1.00 | 0.15 | 0.06 | 0.10 | 1.00 | 0.17 | 0.11 | 0.14 |
| 64 | 1.00 | 0.39 | 0.16 | 0.27 | 1.00 | 0.48 | 0.28 | 0.34 |
| 128 | 1.00 | **0.91** | 0.43 | 0.62 | 1.00 | **1.10** | 0.63 | **0.73** |
| 256 | 1.00 | 1.88 | **1.02** | **1.24** | 1.00 | 2.10 | **1.22** | 1.38 |
| 512 | 1.00 | 3.29 | 2.01 | 2.05 | 1.00 | 3.33 | 2.06 | 2.26 |
| 1024 | 1.00 | 4.89 | 3.26 | 2.95 | 1.00 | 4.54 | 3.03 | 3.20 |

In Table 12 we perform ablation on the value $K$ of nearest neighbors in Equation (4). Although for DAM (Liu et al., 2021) $K$ is set to 10, we found that the value of $K$=128 is generally good for bias mitigation with DenseFace.

Table 13: Ablation study on angular margin, RFW, NIST protocol, FPR @ similarity (Caucasian FPR = $10^{-3}$), $\times 10^3$ scale, the closer to 1 the better

| $m$ | Cauc. | Afr. | Asian | Ind. | Cauc. | Afr. | Asian | Ind. |
|---|---|---|---|---|---|---|---|---|
| | CosFace-R50-Glint360K | | | | AdaFace-R50-WebFace4M | | | |
| $\pi/4$ | 1.00 | **1.04** | 2.29 | 1.92 | 1.00 | 2.99 | 2.10 | 2.23 |
| $\pi/3$ | 1.00 | 0.40 | **0.71** | **0.92** | 1.00 | **1.10** | **0.63** | **0.73** |
| $5\pi/12$ | 1.00 | 0.06 | 0.04 | 0.20 | 1.00 | 0.30 | 0.21 | 0.25 |

In Table 13 we perform ablation on the value of margin $m$ in Equation (6). We found that the value $m = \frac{\pi}{3}$ is optimal for each considered model. If margin is too small it is not enough to deal with orthogonality of the embeddings of the anchor set. If it becomes too big it starts to saturate $f(x)$ in Equation (7) to 1.

## F   MULTI-RACIAL TEST

We also investigated impact of DenseFace on face verification accuracy on multi-racial test that contains all racial groups in one dataset. We combined all images from the RFW dataset into a new test set and performed pairwise comparisons all-vs-all. Multi-racial test contains 40607 images with approximately 57 thousands of positive and 824 millions of negative pairs. Results for the strongest model AdaFace-R100-WebFace12M are presented in Table 14. DenseFace and DenseFace[†] consistently improve results over cosine similarity baseline.

Table 14: Multi-racial test, RFW, TPR @ FPR (%)

| Methods | TPR @ FPR=$10^{-6}$ | TPR @ FPR=$10^{-5}$ |
|---|---|---|
| AdaFace-R100-WebFace12M | | |
| Cosine | 94.72 | 97.84 |
| DenseFace | **95.29** | **98.04** |
| DenseFace[†] | 95.15 | 97.95 |

## G   COMPUTATIONAL COST ANALYSIS AND LIMITATIONS

We have evaluated the computational cost of local density computation with and without the use of the anchor set and compared the results with the Cosine baseline .We use an NVIDIA A100 GPU with batch size 8. As shown in Table 15, the proposed regression network in DenseFace[†] does not add any additional complexity, either in latency or memory compared to baseline.

Table 15: Computational cost in terms of latency and memory. Cosine implies backbone inference to obtain embeddings. DenseFace and DenseFace$^\dagger$ is for the inference and local density estimation based on anchor set or regression network, respectively.

| Methods | Memory, MB | Latency, ms |
|---|---|---|
| Cosine | 248.52 | 17.71 |
| DenseFace | 355.23 (+ 42.93%) | 102.63 (+ 479.54%) |
| DenseFace$^\dagger$ | 249.04 (+ 0.2%) | 18.02 (+ 1.75%) |

We also measured the matching speed according to Equation (9) (see Table 16). We emphasize that the first two terms in Equation (9) could be computed during the local density calculation phase along with the extraction of face embeddings. Also, operations in the third term such as the weighted sum of two embeddings, taking its norm, and calculating the modified Bessel function of the first kind may result in sensible computational cost. To minimize the cost caused by norm operation, we appeal to Numba library Lam et al. (2015) that allows to translate Python code into highly optimized machine code at runtime. We use look up table (LUT) data structure to replace the modified Bessel function of the first kind to proceed with the cost minimization, which results in 40 MB memory trade off to store kappa values and their corresponding function values obtained on RFW and RB-WebFace datasets. The introduced techniques affect neither bias nor verification performance. For time cost measurement, we use an AMD EPYC 7702 64-Core CPU.

Table 16: Cosine and DenseFace matching time cost. Each method is tested on 10k iterations of randomly generated unit vectors, resulting in the average and standard deviation time.

| Methods | Avg. time ratio | Avg. ms | Std. ms |
|---|---|---|---|
| Cosine | - | 0.029 | 0.00346 |
| SCF Li et al. (2021) | 2.5x | 0.072 | 0.00322 |
| DenseFace$^\dagger$ | 2.5x | 0.071 | 0.00319 |
| DenseFace$^\dagger$(LUT) | 2.1x | 0.061 | 0.00330 |
| DenseFace$^\dagger$(LUT + Numba) | 1.5x | 0.042 | 0.00213 |

## H  ANCHOR SET IDENTITY OVERLAP

We noticed that there could be an overlap between identities from anchor set and regressor DenseFace$^\dagger$ training set. It could result in higher values of $\kappa^{(m)}$ density for such overlapping identities. To prevent this effect, we detected these cases by cosine similarity threshold 0.6 and excluded these identity duplicates from the nearest neighbor sets $P_i^K$.

## I  LOCAL DENSITY AS IMAGE FILTER

As face embedding local density helps to mitigate bias in our DenseFace approach, we tried to use $\kappa^{(m)}$ as image filter that allows to remove some samples from the verification task. Since high values of local density indicate high similarity between different identities in this region, we sorted $\kappa^{(m)}$ values in descending order for all test images and removed samples with the highest density values. We varied the percentage of filtered images and calculated FPR at threshold value that initially corresponded to FPR=0.001 and plotted the Error vs. Reject Curves (ERC). The results are shown in Figure 8. Removal of the samples with high local density helps to reduce FPR for African and Indian groups. Caucasian and Asian false positive errors appear to be less sensitive to estimated local densities.

We also show that it can be useful to remove samples with the highest density on multi-racial test. In Figure 9 we plot ERC curves for FPR at fixed initial threshold (FPR=0.001) and FNR (FNR=1-TPR) at fixed level of FPR = 0.001. So, in order to reduce face verification error, there could be two different approaches of sample filtering: we can reduce FPR at the fixed threshold value, or, we can reduce FNR at the fixed FPR value.

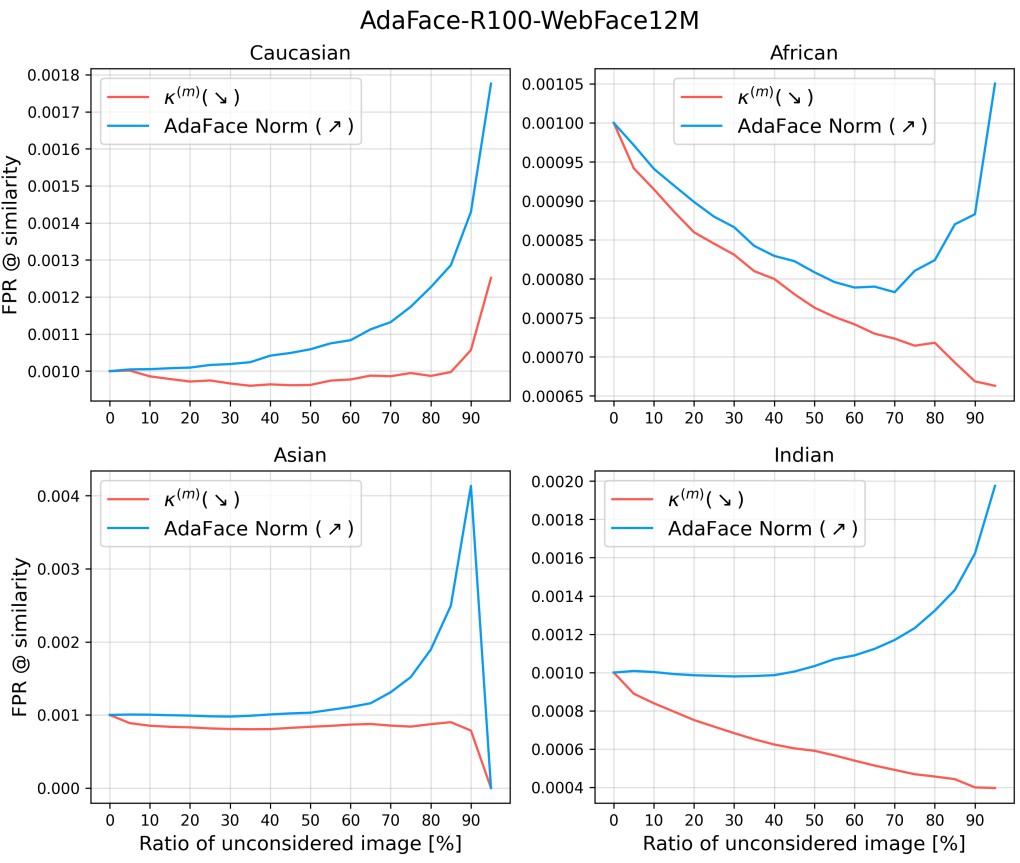

Figure 8: ERC comparison between $\kappa^{(m)}$ density and AdaFace embedding norm as image filter on RFW. The plots show the effect of removing samples with high local density (DenseFace) or low face embedding norm (AdaFace).

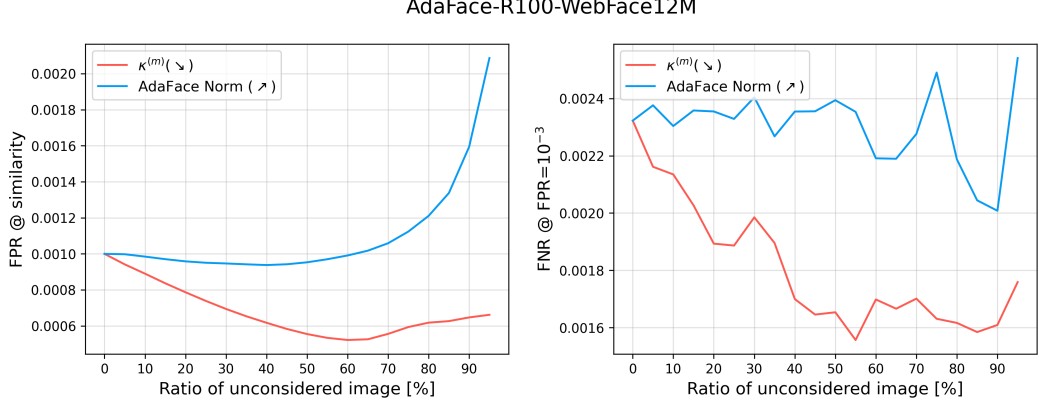

Figure 9: ERC comparison between $\kappa^{(m)}$ density and AdaFace embedding norm as image filter on Multi-racial test, RFW. The plots show the effect of removing samples with high local density (DenseFace) or low face embedding norm (AdaFace).