# OpenReview forum: "DenseFace: Bias Mitigation in Face Recognition via Density-Aware Probabilistic Matching"
_ICLR.cc/2026/Conference — Submitted to ICLR 2026_

### Official Review · Reviewer_zsEY · 2025-10-25

**Soundness:** 1
**Presentation:** 3
**Contribution:** 2
**Rating:** 2
**Confidence:** 5

**Summary:**

The paper proposes a method to mitigate gender and ethnicity bias in face recognition (FR) systems. The key observation is that biased performance across demographic groups correlates with uneven density distributions in the embedding space. Using this, the authors integrate a density-based function into the FR decision process, avoiding the need to retrain the underlying recognition model. The method relies on an auxiliary dataset (Glint360K) to estimate the local embedding density for the images being compared.

**Strengths:**

- The paper’s main idea—using embedding density to modulate similarity scores for fairness without retraining—is original and practically appealing. It provides a promising direction for fairness-aware FR systems.

- The approach is mathematically grounded and builds on a clear geometric intuition regarding embedding distributions.

- Since retraining large FR models is often infeasible, this method could offer a lightweight alternative for improving fairness post-training.

**Weaknesses:**

- Some **basic conceptual errors** undermine confidence in the results: The argument that "standard deviation (std) is inconsistent as a bias metric."

   Table 1 reports verification accuracy, while Figure 5 shows average cosine similarity of impostor (negative) pairs. The authors seem to conflate differences across metrics with inconsistencies in bias measures. Their argument for saying that std is inconsistent is that the std in Table 1 shows the lowest bias for one model, while Figure 5 shows the lowest bias for another model. The apparent contradictions stem from incomplete consideration of the metrics used, not from issues with std. Impostor scores are just one part of the equation, the other part are the genuine scores. It would also be important for the authors to clarify the exact metric used to calculate the verification accuracy.

- The terms intra- and inter-class distributions are used ambiguously, what constitutes a “class” in this context is unclear. While the equations are mathematically clear, the implications of adding a margin to the cosine function are not. It is uncertain whether the resulting function remains a valid probability density function, and it certainly diverges from the von Mises–Fisher (vMF) formulation the authors cite.

- The process for computing and using the anchor set $N_A$ is insufficiently described. The text suggests that an anchor embedding is computed per identity in the selected 54K subset of Glint360K, but it remains unclear how this anchor set is paired or matched with a query image at recognition time. The mention of a “learning-based DenseFace” and the regression network  $\kappa^{(m)}$  are abrupt and never explained.

- Ignoring the predefined RFW test pairs is a poor design choice. These are challenging pairs that were intentionally curated to make cross-demographic differences clearer. Using all possible pairs obscures these differences and prevents comparison between studies.

- The statement “Following Dooley et al. (2024), we question the necessity of demographic balance” contradicts the method’s reliance on a balanced dataset (Glint360K subset) for density estimation. Furthermore, the claim that training on skewed datasets results in lower bias overlooks the effect of group sample sizes and the implicit balance that authors introduce through auxiliary data.

- The paper claims that “values closer to 1 are better” for False Positive Rate (FPR) in Tables 2, 3 and 5, which is technically incorrect. If the authors intend to refer to fairness rather than accuracy, that should be explicitly stated.

- DenseFace is compared only to baselines, but not to other fairness-aware post-processing or threshold adaptation methods that also exploit demographic information (e.g., adaptive thresholds or group calibration techniques).

**Questions:**

1. Why are improvements significant only in tables reporting FPR@similarity (e.g., Tables 2–3, 5–8, 11–12), while other metrics (TPR@FPR, verification accuracy) show marginal or inconsistent gains (e.g., Tables 4, 9–10)?

2. How is the anchor set $N_A$ constructed and used during inference?

3. Can you explain what your modified $\kappa^{(m)}$ represents and how it affects the probability density function?

4. Why not benchmark against existing fairness-oriented FR methods (e.g., adaptive thresholding)?

**Minor question**: In the quantitative analysis, can you explain the rationale behind the sentence _"The presence of other races in nearest neighbors distributions can be explained by the cross-racial effect of bias from other attributes like gender or age"_?

---

### Official Review · Reviewer_Ps9f · 2025-10-31

**Soundness:** 3
**Presentation:** 3
**Contribution:** 3
**Rating:** 6
**Confidence:** 4

**Summary:**

The paper proposes **DenseFace**, a _post-training_ bias-mitigation method for face recognition that **does not retrain the backbone**. The key idea is to treat each embedding as a **von Mises–Fisher (vMF)** distribution on the hypersphere and make the _matching score_ depend on **local inter-class density** rather than raw cosine similarity. Concretely: build a **balanced anchor set** of identity means; estimate **local density** for a query by its $K$-NN neighbors in the anchor set; apply a novel **angular-margin “local distortion”** so that near-orthogonal anchors still yield informative densities; then compute a **probabilistic mutual likelihood** score $S^{(m)}_{ij}$ between two vMFs (closed form via the vMF normalizer), effectively **down-weighting matches in dense regions** and **up-weighting** in sparse ones. Using NIST-style **FPR@threshold fixed on the Caucasian cohort**, DenseFace substantially reduces cross-group FPR gaps on **RFW** and **RB-WebFace** across **CosFace/AdaFace** models—while **preserving TPR**—and a tiny **learned regressor** for $\kappa^{(m)}$ further improves fairness with negligible overhead.

**Strengths:**

* **No retraining, minimal overhead.** Works atop strong backbones (AdaFace, CosFace) trained on large skewed datasets; DenseFace† adds ~0.2% params / ~1.75% MACs.

* **Metric realism.** A persuasive case for **FPR@common threshold** vs. per-group thresholds/ROC ranges; demonstrates counter-intuitive cases where Std-of-accuracy misorders model bias.

* **Substantive fairness gains.** RFW (NIST protocol): e.g., African FPR shrinks **8.82→0.91×10⁻³** (CosFace-R50), Asian **7.69→0.43×10⁻³**; RB-WebFace shows similar drops—all **without TPR loss** at 1e-3/1e-4.

* **Interpretable link.** Higher inter-class densities ($\kappa^{(m)}$) correlate with worse FPRs; neighbor race histograms show anchor K-NNs primarily within the same race, explaining **label-free** operation at test time given a balanced anchor.

**Weaknesses:**

* **Anchor dependence / hidden labels.** Building a _balanced_ anchor set relies on **race/gender classifiers** over Glint360K; fairness of DenseFace may inherit their errors/biases (even if test-time is label-free). Please quantify sensitivity to misclassified anchors.

* **Hyperparameter sensitivity.** Results could shift with $K$ and **margin $m$** in the local distortion; ablation curves for FPR parity vs. $K,m$ are missing.

* **Computation vs. scale.** Although JIT/look-ups help, **1.5×** matching overhead vs. cosine is non-trivial at web-scale identification; more profiling under large gallery sizes would help.

* **Scope of robustness.** The paper focuses on race/gender bias; no study on **age/quality/blur/occlusion** strata where density patterns might differ—useful for deployment claims.

* **Fairness trade-offs elsewhere.** With a **single** threshold set on the Caucasian cohort, what happens to **false negatives** across groups? TPR is tabulated, but more granularity (e.g., DET sweeps around the operating point) would clarify trade-offs.

* **Missing Related Works.** For example, but not limited to,
   - Robinson, J. P., Livitz, G., Henon, Y., Qin, C., Fu, Y., & Timoner, S. (2020). Face recognition: too bias, or not too bias?. In Proceedings of the ieee/cvf conference on computer vision and pattern recognition workshops (pp. 0-1).
There are several spots where this could extend the story in Section 2.3 with minimal extra text, providing a more complete overall picture. There are others, too, but including the ones directly related is essential for sound research and complete storytelling.

**Questions:**

1. **Anchor noise.** If 5–10% of anchor identities are mislabeled demographically (via the pre-classifier), how do the FPR gains change? Any _self-balancing_ strategy without demographic pre-labels?

2. **Sensitivity to $K,m$.** Please provide **FPR ratio curves** vs. $K\in\{32,64,128,256\}$ and margin $m$; also report stability of $\kappa^{(m)}$ under these changes.

3. **Operational profiling.** What is the end-to-end **QPS** impact at gallery sizes 1e6–1e8 for verification/identification? Any **approximate vMF** tricks (e.g., series expansions) beyond LUT/JIT?

4. **Generalization.** Do **age** or **image-quality** cohorts show similar density–FPR coupling? A brief analysis on AgeDB/quality splits would strengthen the story.

5. **Thresholding policy.** If operators fix the threshold on **non-Caucasian** or **pooled** cohorts, do findings persist? Please add a small study varying the _reference_ cohort for threshold selection.


6. **DenseFace† details.** What features feed the $\kappa^{(m)}$ regressor? Only $z$ or also neighborhood stats? Does it learn implicit demographics versus pure density? Ablate input choices.

**Details Of Ethics Concerns:**

The method itself is a post-hoc calibration, but constructing a balanced anchor requires demographic attribute inference over large web-scale datasets. Please ensure licensing/consent for attribute inference and discuss safeguards to prevent the same density signal from being misused for group attribution at inference time.

---

### Official Review · Reviewer_XYSP · 2025-11-02

**Soundness:** 3
**Presentation:** 2
**Contribution:** 2
**Rating:** 2
**Confidence:** 5

**Summary:**

This paper addresses demographic bias in face recognition without requiring retraining (on top solution, following a line of works). The authors propose DenseFace a density-aware probabilistic matching framework that models face embeddings as samples from von Mises-Fisher distributions. They show that the local density of face embeddings correlates with demographic attributes, leading to uneven similarity scores across racial and gender groups. DenseFace estimates these densities using an anchor set and adjusts the similarity scores by expanding or contracting regions of the embedding space. The authors also propose a learning-based variant that predicts local density parameters by a small regression network.

**Strengths:**

- the post training direction is very practical and have shown success previously. This fmily of approaches is very practical.
- It demonstrates strong empirical generality across datasets and model families (at least in the presented metric, which will come to later).

**Weaknesses:**

- The method requires constructing a balanced anchor set, which may itself rely on demographic label partially contradicting the label-free inference claim. This have to be better investigated and discussed in comparison to other training free methods.
- Experiments are primarily on racial bias; gender and age bias analyses are less explored.
- there so much promotion of NIST protocol but even using the wrong metric when talking about it. FPR is not FMR. Additionally, NIST protocol is motivated by having a product with fixed threshold and the need to evaluate it. Works such as this one should aim to understand better what is happening and why is it happening, which a single threshold captured from one demographic is far from being able to present that.
- Even though the paper acknowledge the existance of many methods that present on-the-top bias mitigation solutions, it does not sufficienly theoretically ddifferentiate between itself and these works. It also presents no comparison to these methods experimentally (comparison to one random method in suplimentary material makes this even more suspicious). Remember that beating SOTA is not the goal but presenting knowledge is, and for that you need comparison.
- It is not clear why models based on one single loss are used. Why not measuring generalization on models with diversity of losses.
- Not clear why some points are proved using the claimed to be NIST metric and others use the STD metric. why not both or one of them.
- saying always SOTA FR while just using one loss is a bit problematic.
 -while reproting traditional metrics in table 10 on the SM, why only 3 out of the 4 used models in the paper is not clear.

**Questions:**

1. How are demographic labels obtained for constructing the balanced anchor set, and does this contradict the “label-free” claim?
2. How does DenseFace compare theoretically to other post-training bias mitigation methods? why experiment comparison is not presented?
3. Why are experiments limited to racial bias?
4. Why is the NIST protocol promoted (despite misusing FPR instead of FMR) and that is used for different experimental scenario where the products are fixed and no need to get additional knowledge about them?
6. Why are models trained with only one loss function used, while claiming generality across SOTA FR models?
7. Why are results mixed between NIST metrics and STD bias measures, why not use both or one consistently?
8. Why is the “SOTA” claim made when only a subset of loss functions and models are evaluated?
9. Why does Table 10 in the supplementary include only three of the four models used in the main paper?

---

### Meta-Review · Area_Chair_xYFv · 2026-01-08

**Summary:**

Reviewers agreed that using post-training, density-based methods to reduce bias is a practical and promising idea. However, they raised serious concerns about unclear concepts, incorrect or inconsistent use of metrics, reliance on demographic labels, missing comparisons to other fairness methods, and inconsistent evaluation practices. Because of these issues, reviewers felt that the paper lacks sufficient rigor and clear positioning. As a result, the paper received mostly negative or borderline recommendations and will be rejected from ICLR.

**Reviewer Concerns:**

There is no rebuttal.

**Reviewer Scores:**

There is no rebuttal.

---

### Decision · Program_Chairs · 2026-01-26

Reject